# On Direct Distribution Matching for Adapting Segmentation Networks

**Georg Pichler**[1]                                                   GEORG.PICHLER@TUWIEN.AC.AT
**Jose Dolz**[2]                                                            JOSE.DOLZ@ETSMTL.CA
**Ismail Ben Ayed**[2]                                              ISMAIL.BENAYED@ETSMTL.CA
**Pablo Piantanida**[3]                    PABLO.PIANTANIDA@L2S.CENTRALESUPELEC.FR
[1] *TU Wien, Gusshausstrasse 25/E389, Vienna, Austria*

[2] *ÉTS, 1100 Notre Dame St W, Montreal, Canada*

[3] *CentraleSupélec-CNRS-Université Paris Sud, 3 rue Joliot-Curie, Gif-sur-Yvette, France*

## Abstract

Minimization of distribution matching losses is a principled approach to domain adaptation in the context of image classification. However, it is largely overlooked in adapting segmentation networks, which is currently dominated by adversarial models. We propose a class of loss functions, which encourage direct kernel density matching in the network-output space, up to some geometric transformations computed from unlabeled inputs. Rather than using an intermediate domain discriminator, our direct approach unifies distribution matching and segmentation in a single loss. Therefore, it simplifies segmentation adaptation by avoiding extra adversarial steps, while improving quality, stability and efficiency of training. We juxtapose our approach to state-of-the-art segmentation adaptation via adversarial training in the network-output space. In the challenging task of adapting brain segmentation across different magnetic resonance imaging (MRI) modalities, our approach achieves significantly better results both in terms of accuracy and stability.

**Keywords:** domain adaptation, unsupervised domain adaptation, semantic segmentation, direct distribution matching

## 1. Introduction

Semantic segmentation is of pivotal importance towards high-level understanding of image content, which is useful in a breadth of application areas, from autonomous driving to health care, for instance. Particularly, in medical imaging, segmentation facilitates clinical tasks, including disease diagnosis, treatment and follow-up, among others. Modern medical segmentation approaches rely on deep learning techniques, which have demonstrated outstanding performance in a breadth of applications (Dolz et al., 2018a,b; Litjens et al., 2017). Despite their success, generalization of trained models to new scenarios is hampered if the gap between data distributions across domains is large. A trivial solution to address this issue would be to re-annotate images from different domains and re-train or fine-tune the deep models. Nevertheless, obtaining such massive amounts of labeled data is a cumbersome process which, for some applications, may require user expertise, resulting in a prohibitive and unrealistic approach.

To tackle this problem, unsupervised domain adaptation (UDA) techniques have been widely investigated. These methods aim at learning robust classifiersin the presence of a *shift* between

source and target distributions when the target data is unlabeled. In this scenario, the goal is typically to minimize the discrepancy between distributions across domains at the input (Bousmalis et al., 2017; Chen et al., 2018a; Hoffman et al., 2018; Russo et al., 2018; Sankaranarayanan and Balaji, 2018; Wu et al., 2018) or intermediate-feature level (Ganin and Lempitsky, 2015; Ghifary et al., 2016; Kamnitsas et al., 2017; Liu et al., 2018; Long et al., 2015, 2016; Tzeng et al., 2017), while leveraging labeled source examples to retain discriminative power on the feature space. Generative techniques either operate on a pixel-level (Bousmalis et al., 2017; Chen et al., 2018a; Russo et al., 2018; Shrivastava et al., 2017; Zhang et al., 2018a) or in feature space (Dou et al., 2018b; Ganin and Lempitsky, 2015; Kamnitsas et al., 2017; Long et al., 2015; Tzeng et al., 2017) and align the image appearance between domains, so that the target data "style" is transferred to source data, or vice-versa. Then, supervised learning is performed with the newly generated synthetic data. A downside of these approaches is that they perform satisfactorily only for small images and narrow domain shifts, which limits their applicability. Within the current paradigm of learning domain-invariant representations, domain adversarial training (Ganin and Lempitsky, 2015; Tzeng et al., 2017) and maximum mean discrepancy (MMD) (Long et al., 2015; Sun and Saenko, 2016; Yan et al., 2017) have become very popular choices.

For semantic segmentation problems, adversarial training models (Goodfellow et al., 2014) are currently dominating the literature (Chen et al., 2017, 2018b; Dou et al., 2018a; Kamnitsas et al., 2017; Hoffman et al., 2016; Hong et al., 2018; Saito et al., 2018; Tsai et al., 2018; Vu et al., 2019). Such models alternate the training of two networks: a discriminator that learns a decision boundary between source and target features and a segmentation network that uses the learned decision boundary to match the feature distributions across domains. Some other approaches rely on generative networks, which yield target images conditioned on the source, or vice-versa, aligning both domains at the pixel level (Cai et al., 2019; Huo et al., 2018; Murez et al., 2018; Sankaranarayanan and Balaji, 2018; Zhang et al., 2018b; Zhao et al., 2019).

While adversarial training achieved outstanding performances in image classification, our numerical evidence and intuition suggest that it may not be suitable for segmentation tasks to the same degree. First, learning a discriminator boundary for a segmentation task is much more complex as the label space is exponentially large. Intuitively, a high dimensional label space implies that the discriminator boundary can be very complicated and thus hard to learn. Therefore, as we will see later in our experiments, alternating both adversarial and prediction tasks in segmentation might cause more significant training instabilities than in image classification tasks. Moreover, it is more unlikely that source and target domain share the same multi-level feature representations if the label space is high dimensional.

While the inputs can differ significantly from one domain to another, the output (label) space in semantic segmentation conveys very rich information related to the spatial layout and local context, which is shared across domains. Inspired by this observation, Tsai *et at.*(Tsai et al., 2018) proposed adversarial training in the output (softmax segmentation) space, achieving better performance than features-matching approaches on the Cityscapes dataset. Leveraging this information is even more meaningful in medical images, where label (output) statistics remain domain-independent, despite significant differences in image inputs across domains. Nevertheless, following the trend in UDA approaches for natural image segmentation, adversarial learning has become the *de facto* choice in medical image segmentation (Chen et al., 2019; Dou et al., 2018a; Gholami et al., 2018; Javanmardi and Tasdizen, 2018; Kamnitsas et al., 2017; Zhang et al., 2018a; Zhao et al., 2019). It is worth mentioning that some recent natural image segmentation works (Zhang et al., 2017; Zou et al.,

2018) pointed out that adversarial models for classification do not translate well to segmentation. These studies showed that similar or better performances can be achieved by other alternatives.

Here, we propose a simple, easily trainable approach to UDA, that can be applied in cases where the underlying (latent) ground truth is identical for source and target domains, up to some geometric transformations of unlabeled images. While unrealistic for natural images, this can easily be achieved in medical imaging, e.g., by obtaining separate scans of one patient with different imaging methods or by applying multi-modal registration algorithms to unlabeled image pairs. The class of loss functions we propose encourages direct density matching in the network's output space. It follows the principle of Minimization of distribution matching losses, a principled approach to domain adaptation (DA) in the context of image classification, e.g., MMD (Long et al., 2015; Sun and Saenko, 2016; Yan et al., 2017). Rather than using an intermediate domain discriminator, our direct approach unifies distribution matching and segmentation in a single loss. Therefore, it simplifies segmentation adaptation by avoiding extra adversarial steps, while improving quality, stability and efficiency of training. We compare our approach to the state-of-art segmentation method in (Tsai et al., 2018). In the challenging task of adapting brain segmentation across different magnetic resonance imaging (MRI) modalities, our approach achieves significantly better performance than adversarial output adaption, both in terms of accuracy and stability. We also investigate experimentally the sensitivity of our approach to the alignment of unlabeled image pairs.

## 2. Formulation

Consider an unsupervised domain-adaptation setting with two distinct subsets: $\mathcal{L} = \{(X_i, Y_i)\}_{i=1,\ldots,n}$ contains labeled source-domain images $X_i$ and the corresponding ground-truth segmentations $Y_i$, and $\mathcal{U} = \{(X_i, X_i')\}_{i=n+1,\ldots,n+m}$ contains *unlabeled* image pairs, each involving a source image $X_i$ and a target image $X_i'$. For each labeled source image $X_i : \Omega \subset \mathbb{Z}^{2,3} \to \mathbb{R}$, $i = 1, \ldots, n$, the ground-truth labeling $Y_i \in \{0, 1\}^{L \times |\Omega|}$ is a matrix whose columns are binary vectors, encoding the assignment of pixel $p \in \Omega$ to one of $L$ classes (segmentation regions): $\mathbf{y}_i(p) = (y_i(1, p), \ldots, y_i(L, p)) \in \{0, 1\}^L$, where $y_i(l, p) = 1$ if and only if label $l$ is assigned to pixel $p$ of the $i$-th image. For any image $X$, let $\mathbf{s}_\theta(p, X) = (s_\theta(1, p, X), \ldots, s_\theta(L, p, X)) \in [0, 1]^L$ denote the probability vector of softmax outputs for pixel $p$, with $\theta$ the trainable parameters of the network. For the sake of simplicity, we will omit the subscript $\theta$ in the following.

We propose to minimize the following loss function:

$$\mathcal{F}(\theta) = \sum_{i=1}^{n} \sum_{p \in \Omega} \mathcal{H}(\mathbf{y}_i(p), \mathbf{s}(p, X_i)) + \lambda \sum_{i=n+1}^{n+m} \sum_{p \in \Omega} \mathcal{D}(\mathbf{s}(p, X_i), \mathbf{s}(p, \mathcal{T}(X_i'))), \qquad (1)$$

where

- $\mathcal{D}(\mathbf{s}, \mathbf{s}')$ evaluates the discrepancy between two probability distributions $\mathbf{s}$ and $\mathbf{s}'$, e.g., Kullback-Leibler (KL) divergence $\mathcal{D}_{\mathrm{KL}}(\mathbf{s}, \mathbf{s}') = \mathbf{s}^T \ln \frac{\mathbf{s}}{\mathbf{s}'}$, where superscript $\cdot^T$ denotes transposition.

- $\mathcal{H}$ denotes standard cross-entropy loss for labeled source-domain images: $\mathcal{H}(\mathbf{y}, \mathbf{s}) = \mathcal{D}_{\mathrm{KL}}(\mathbf{y}, \mathbf{s})$.

- $\lambda$ is a non-negative multiplier.

- $\mathcal{T}$ could be simply identity if unlabeled images $X_i$ and $X_i'$ are aligned, e.g., by acquisition[1]. Also, $\mathcal{T}$ could be a geometric transformation, which aligns pairs of unlabeled images, for instance, using a standard automatic cross-modality registration algorithm (Oliveira and Tavares, 2014).

The first term in our model (1) is the usual cross-entropy loss of a semantic segmentation problem on the source domain, while the second term, which is based on unlabeled image pairs, encourages the network outputs (softmax segmentations) in the target domain to closely match those in the source domain. In fact, when $\mathcal{D}$ corresponds to some kernel function, i.e., $\mathcal{D}(\cdot, \cdot) = -\mathcal{K}(.,.)$, the summation over pixels in the second term of (1) can be expressed in terms of a kernel $\tilde{\mathcal{K}}$ between two softmax segmentations in $\{0, 1\}^{L \times |\Omega|}$:

$$- \tilde{\mathcal{K}}\big(S(X_i), S(\mathcal{T}(X_i'))\big) = \sum_{p \in \Omega} \mathcal{D}(\mathbf{s}(p, X_i), \mathbf{s}(p, \mathcal{T}(X_i'))) \tag{2}$$

with $S(X) \in \{0, 1\}^{L \times |\Omega|}$ denoting the matrix whose columns are the softmax outputs at each pixel, i.e., probability vectors $\mathbf{s}(p, X)$. Now, notice that the *kernel density estimate (KDE)*[2] of the distribution of source-domain softmax segmentations, i.e., the network outputs in $\{0, 1\}^{L \times |\Omega|}$, can be written as follows: $\mathcal{P}(S(X)) \propto \sum_{i=n+1}^{m} \tilde{\mathcal{K}}(S(X_i), S(X)), \forall X$. Therefore, by maximizing these source density estimates at target-domain segmentations, we directly match the distributions of the source and target domains in the network-output space. This amounts to minimizing the following direct distribution-matching loss:

$$- \sum_{j=n+1}^{m} \mathcal{P}(S(\mathcal{T}(X_j'))) = - \sum_{i,j=n+1}^{m} \tilde{\mathcal{K}}(S(X_i), S(\mathcal{T}(X_j'))). \tag{3}$$

Clearly, from the expression of kernel $\tilde{\mathcal{K}}$ in (2), the second term in our loss in (1) can be viewed as an approximation of (3) based on a subset of pairwise matching kernels. Therefore, our loss in (1) encourages direct density matching in the network-output space.

Fig. 1 highlights the conceptual differences between our direct matching (Fig. 1(c)) and the state-of-art adversarial method in (Tsai et al., 2018), which pursues a two-step adversarial learning in the network-output space (Fig. 1(a) and 1(b)), so as to achieve the same goal as our loss: matching the source and target distributions of label predictions. The model in (Tsai et al., 2018) alternates the training of two networks: a discriminator, which learns to distinguish between source and target outputs; and a segmentation network, which is trained using the discriminator. The discriminator is used to encourage the target outputs to be similar to those of the source domain. Rather than using an intermediate domain discriminator, our direct method unifies distribution matching and segmentation in a single loss. Therefore, it simplifies segmentation adaptation by avoiding extra adversarial steps, while improving both the quality, stability and efficiency of training. While adversarial training achieved outstanding performances in image classification, our numerical evidence and intuition suggest that it may not be suitable for segmentation, in which case learning a discriminator boundary is much more complex as it solves for predictions in an exponentially large label space. In fact, intuitively, a large label space implies large spaces of possible solutions for

---

1. In some practical scenarios, images from different modalities are aligned when acquired at the same time.
2. KDEs are also commonly referred to as Parzen window estimates.

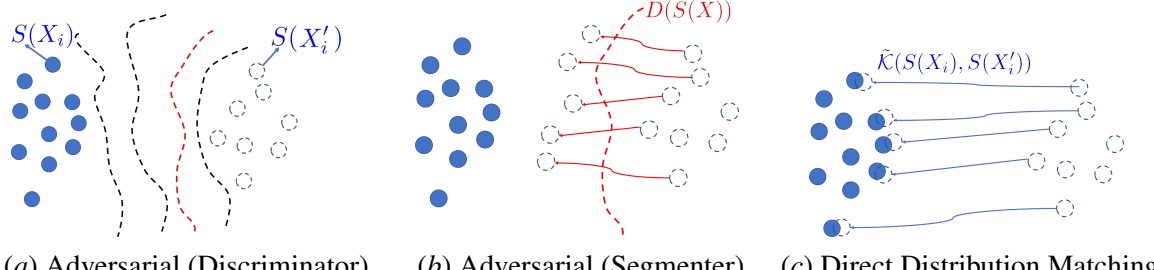

(*a*) Adversarial (Discriminator)  (*b*) Adversarial (Segmenter)  (*c*) Direct Distribution Matching

Figure 1: A conceptual juxtaposition of adversarial training in the network-output space (Tsai et al., 2018) (Fig. 1(*a*) and 1(*b*)) and our direct density matching (Fig. 1(*c*)). The data points in the figure depict networks outputs (softmax segmentations), with the blue points corresponding to the source and dashed points to the target.

discriminator boundaries and target predictions, both of which are latent; see dashed boundaries and data points in Fig. 1(*a*). Alternating both adversarial and prediction tasks in segmentation can cause more significant instabilities than in image classification tasks, as we will see later in our experiments.

Another important difference between our approach and adversarial training is that we account for the fact that target and source data have a common ground truth in the label space, up to some geometric transformation. Such prior information is very common and useful in medical imaging problems, but adversarial approaches do not have mechanisms to take advantage of it.

## 3. Experiments

We evaluated our approach extensively on the challenging task of brain tissue segmentation in MRI scans, and compared the performances to the state-of-the-art method in (Tsai et al., 2018).

### 3.1. Experimental details

**Datasets:** We performed numerical studies on two public segmentation benchmarks: MR-BrainS2013 (Mendrik et al., 2015) and iSEG2017 (Wang et al., 2019). The MRBrainS dataset contains 5 labeled and 15 unlabeled scans of adult brains. The iSEG dataset is composed of 10 labeled and 13 unlabeled infant brain scans. We tested our domain adaptation on the T1 and T2-FLAIR modalities of MRBrainS and the T1 and T2 modalities of iSEG. The task consists of segmenting the white matter (WM), gray matter (GM) and cerebrospinal fluid (CSF). The original T2 images from iSEG were resampled into an isotropic $1 \times 1 \times 1\mathrm{mm}^3$ resolution, and then aligned onto their corresponding T1 images with a simple affine registration method. The sequences from the MRBrainS Challenge were aligned by rigid registration, using Elastix (Klein et al., 2010).

**Training:** The data $(\mathcal{L}, \mathcal{U})$ consists of two distinct subsets. $\mathcal{L} = \{(X_1, Y_1), \ldots, (X_n, Y_n)\}$ is the labeled subset, which contains images $X_i$ from the source domain with their corresponding ground truth $Y_i$. Unlabeled subset $\mathcal{U} = \{(X_{n+1}, X'_{n+1}), \ldots, (X_{n+m}, X'_{n+m})\}$ contains pairs of aligned source and target data, respectively, without a ground-truth. In our experiments, we found that the choice of distance functions $\mathcal{D}$ does not significantly alter performances. If not mentioned otherwise, we used KL divergence $\mathcal{D}(\mathbf{s}, \mathbf{s}') = \mathbf{s}^T \ln \frac{\mathbf{s}}{\mathbf{s}'}$ and the multiplier $\lambda = 0.01$.

Due to the limited size of the training set, we employed a leave-one-out-cross-validation strategy, where only one image was used for testing/evaluation, leaving the remaining images for training. We used four of the five labeled scans in the MRBrainS2013 dataset as samples in $\mathcal{L}$. The one remaining scan was used for evaluation. As the iSEG dataset contains more labeled scans, we opted to use 8 scans for training in $\mathcal{L}$ and one scan for testing and evaluation, respectively. Furthermore, all unlabeled scans, i.e., 15 and 13 in the MRBrainS2013 and iSEG datasets, respectively, are used in $\mathcal{U}$ to compute the unsupervised term in (1). Each experiment was performed three times with different evaluation/testing data splits and the average as well as the empirical standard deviation reported subsequently were computed over these three runs.

**Baselines and comparisons:** In order to evaluate the impact of the adaptation approaches, we trained the segmentation network in a supervised manner on the source and target data, providing a lower and upper bound for the UDA results. While the network trained on source images is referred to as *no adaptation*, the network trained on the target domain is referred to as the *oracle*. In addition, we compare the proposed approach with the adversarial method proposed in (Tsai et al., 2018). For a fair comparison, we used the same segmentation network for the proposed and the adversarial approach. For simplicity, we chose the "single-level" strategy, performing DA only on the output layer. We used the same discriminator model as (Tsai et al., 2018). The Lagrange multiplier for training the segmentation network was chosen to be $\lambda_{\mathrm{adv}} = 0.1$. Although AdaptSegNet does not utilize the fact that source and target data are aligned in $\mathcal{U}$, we nevertheless trained the discriminator with these aligned pairs. Subsequent runs indeed revealed that this does not have an impact on the performance of AdaptSegNet.

**Implementation details:** We used a slightly modified U-Net (Ronneberger et al., 2015) for the segmentation task, operating on 2D slices. Particularly, the employed network follows the original implementation (Ronneberger et al., 2015), but the depth is reduced by one, i.e., max-pool is performed only three times instead of four. We used ReLU activation functions and did not include dropout, to avoid any regularization that does not originate from our proposed DA strategy. To obtain 2D input, the 3D images are sliced along the z-axis. However, Dice coefficients are computed on the 3D scans. The implementation was done in TensorFlow, and the experiments were run on a server equipped with a NVidia Titan V GPU with 12 GB memory. For all networks, we employed the Adam (Kingma and Ba, 2015) optimizer with learning rate lr = 0.0001 and a batch size of 32. We performed fully supervised pre-training for 200 epochs on the source domain data. Subsequently, we trained for 800 epochs with the full loss (1), totaling 1000 training epochs. The code is publicly available at `https://github.com/g-pichler/DDMSegNet`.

**Evaluation:** We resorted to the common Dice coefficient, widely employed in medical image segmentation, to compare quantitatively the performances of the different methods. When using the iSEG dataset, the mean Dice coefficient on the test scan was used to determine the best model during training. We then report the performance of this model on the evaluation sample. Due to the limited size of the MRBrainS2013 dataset, here, the testing and evaluation sets are identical.

We report Dice coefficients in percent and when comparing the performance of two models, we refer to the absolute difference in percentage points (pp).

### 3.2. Results

Table 1 reports the class-specific and mean Dice coefficients in percent. Looking at the results achieved by the oracle, one can observe that, without adaptation, the performance drops dramat-

| | | | Mean Dice | | | |
|---|---|---|---|---|---|---|
| | | | Oracle | No adaptation | AdaptSegNet | Proposed |
| Source | Target | | Target⟶Target | Source⟶Target | Source⟶Target | Source⟶Target |
| MRB (T1) | MRB (T2-FLAIR) | GM | 77.04 ± 1.27 | 48.81 ± 4.29 | 54.18 ± 9.25 | 76.29 ± 0.80 |
| | | WM | 79.64 ± 4.07 | 12.52 ± 10.82 | 56.54 ± 9.11 | 78.88 ± 3.12 |
| | | CSF | 75.37 ± 1.10 | 54.41 ± 6.88 | 59.12 ± 7.05 | 73.13 ± 2.76 |
| | | **Mean** | **77.35 ± 1.35** | **38.58 ± 1.14** | **56.62 ± 8.02** | **76.10 ± 0.45** |
| MRB (T2-FLAIR) | MRB (T1) | GM | 84.32 ± 0.40 | 10.32 ± 1.89 | 71.87 ± 3.91 | 82.88 ± 0.31 |
| | | WM | 87.46 ± 2.11 | 8.44 ± 6.49 | 77.02 ± 3.14 | 86.43 ± 2.56 |
| | | CSF | 82.36 ± 0.92 | 41.97 ± 6.56 | 70.76 ± 0.93 | 77.99 ± 0.83 |
| | | **Mean** | **84.71 ± 0.98** | **20.25 ± 3.54** | **73.22 ± 2.16** | **82.43 ± 0.50** |
| iSEG (T1) | iSEG (T2) | GM | 75.88 ± 0.85 | 36.48 ± 25.89 | 62.70 ± 5.27 | 73.45 ± 0.94 |
| | | WM | 69.42 ± 0.68 | 31.25 ± 5.20 | 55.16 ± 4.88 | 64.23 ± 0.39 |
| | | CSF | 85.36 ± 1.21 | 48.37 ± 2.97 | 72.25 ± 8.65 | 84.83 ± 1.35 |
| | | **Mean** | **76.89 ± 0.67** | **38.70 ± 10.46** | **63.37 ± 6.25** | **74.17 ± 0.78** |
| iSEG (T2) | iSEG (T1) | GM | 81.15 ± 0.33 | 69.38 ± 1.28 | 70.82 ± 1.78 | 77.34 ± 0.27 |
| | | WM | 76.21 ± 1.53 | 58.24 ± 0.51 | 61.51 ± 3.34 | 68.99 ± 2.77 |
| | | CSF | 89.47 ± 0.97 | 71.17 ± 0.78 | 78.00 ± 4.02 | 87.34 ± 0.85 |
| | | **Mean** | **82.28 ± 0.88** | **66.26 ± 0.53** | **70.11 ± 3.00** | **77.89 ± 1.15** |

Table 1: DA results on MRBrainS and iSEG dataset, showing the Dice coefficient over the three classes (i.e., GM, WM and CSF) as well as the mean. Coefficients are given in percent.

ically, particularly for WM. The adversarial adaptation strategy proposed in (Tsai et al., 2018), AdaptSegNet, is able to infer target domain information during learning and to recovers segmentation performance. For example, when shifting from T1 to T2, AdaptSegNet improves the mean performance by at least 18pp in comparison to *no adaptation*, in both MRBrainS and iSEG images. Despite this improvement, there is still a considerable gap of at least 13.5pp compared to the oracle. On the other hand, the increased performance achieved by our method is more pronounced, getting closer to the performance of the oracle. Particularly, in all the four settings, differences with respect to training the network on target images and our method are in the range between $1.2pp - 4.4pp$. Furthermore, in most cases, the standard deviation is largely decreased by employing the proposed approach rather than the adversarial method. Another interesting finding when independently analyzing the class-specific results is that the proposed method reliably follows the behavior of the oracle. For each of the four analyzed settings, the class segmentation rank for both oracle and proposed approach remains the same.

Qualitative results of these models are depicted in Figure 2. Specifically, cross-sectional 2D MRI scans of two given patients are shown, for both source and target domains, along with the corresponding ground truth and segmentation masks obtained by the different models. We can observe that if no adaptation method is applied, the model trained on the source domain completely fails to segment the target image. Including an adaptation adversarial module visually improves the segmentation, which aligns with the numerical values reported in Table 1. Having a closer look to the AdaptSegNet segmentation, we observe that while the CSF (in brown) seems to correlate with the ground truth, both WM and GM (in yellow and green, respectively) only capture global information, being imprecise in local details. This can be due to the fact that appearance of this

particular structure remains similar across domains, whereas intensity distribution of white and GM highly differ between source and target domains. Indeed, this observation also holds for the *no adaptation* setting, where CSF segmentation obtains the best performance for DA on MRBrainS. Contrary, the proposed direct distribution matching method is able to correctly capture differences between images, satisfactorily adapting both domains.

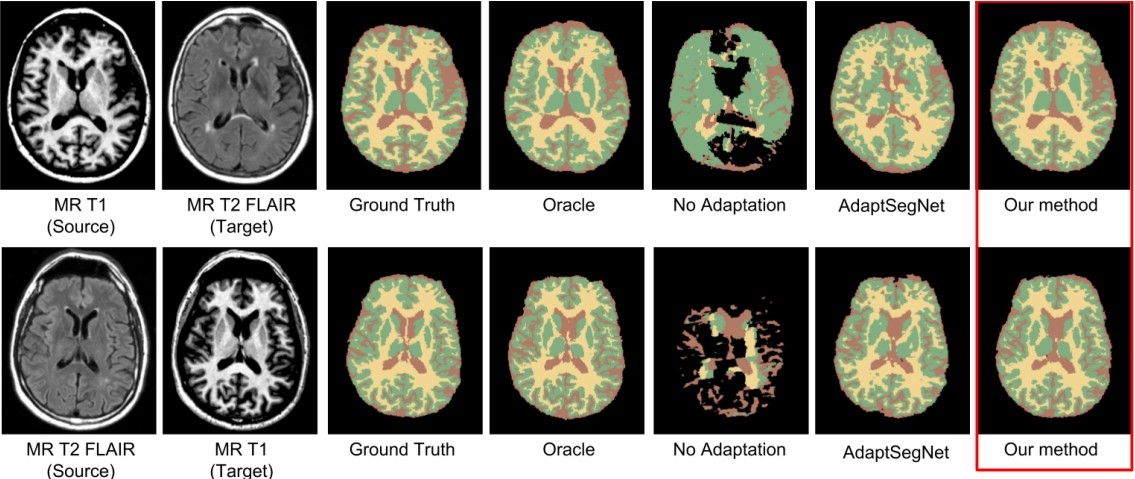

Figure 2: Visual results for two MRBrainS subjects achieved by the different models in the case of adapting a T1-trained model to T2-FLAIR images (*top*), and a T2-FLAIR-trained model to T1 images (*bottom*). These images were randomly selected from one of the three runs.

### 3.2.1. SENSITIVITY TO IMAGE DISALIGNMENT.

Our proposed method assumes perfectly aligned images between the source and the target domain in the unlabeled training set $\mathcal{U}$. In order to test the sensitivity of our approach to a violation of this assumption of alignment between $X$ and $X'$, we should pair scans of different individuals in $\mathcal{U}$. As the datasets are small, instead, we deliberately shuffled the unsupervised pairs using a cyclic shift, and then performed our experiments with the modified unlabeled training data $\mathcal{U} = \{(X_{n+1}, X'_{n+2}), \ldots, (X_{n+m-1}, X'_{n+m}), (X_{n+m}, X'_{n+1})\}$. However, in order to avoid a misalignment due to the imaging procedure, we did perform a 3D affine registration using the SimpleITK software package, registering $X'_{n+i+1}$ to $X_{n+i}$ using mutual information (Mattes et al., 2003) as optimization metric.

The results are detailed in Table 2. When adapting from T1 to T2, the proposed approach achieves similar results than the adversarial method, even offering a slight increase of 0.6pp and 5pp in the iSEG and MRBrainS datasets, respectively. On the iSEG dataset neither method substantially outperforms the *no adaptation* strategy, both AdaptSegNet and the proposed method being within 1.6pp. Only when adapting from T2 to T1 on the MRBrainS dataset, while still improving substantially upon *no adaptation*, AdaptSegNet outperforms the proposed approach by 11.3pp.

While AdaptSegNet does not leverage the alignment between images, while our proposed approach is built upon the assumption of perfect image alignment. However, the data in Table 2 suggest that yet the proposed approach might sill be useful if the alignment between the domains is not perfect and, e.g., achieved by a pre-registration step.

|  | Oracle | No adaptation | AdaptSegNet | Proposed |
|---|---|---|---|---|
| MRBrainS, T1 $\rightarrow$ T2 | $77.35 \pm 1.35$ | $38.58 \pm 1.14$ | $51.50 \pm 6.48$ | $56.52 \pm 3.12$ |
| MRBrainS, T2 $\rightarrow$ T1 | $84.71 \pm 0.98$ | $20.25 \pm 3.54$ | $68.46 \pm 1.17$ | $57.17 \pm 1.00$ |
| iSEG, T1 $\rightarrow$ T2 | $76.89 \pm 0.67$ | $38.70 \pm 10.46$ | $54.40 \pm 3.94$ | $54.99 \pm 1.24$ |
| iSEG, T2 $\rightarrow$ T1 | $82.28 \pm 0.88$ | $66.26 \pm 0.53$ | $67.86 \pm 1.10$ | $66.24 \pm 0.94$ |

Table 2: Mean Dice coefficient in percent when there is misalignment between the images, but an affine registration is performed prior to training.

**Training stability:** In addition to segmentation performance, we juxtaposed our method to the adversarial approach in terms of learning stability. Fig. 3 depicts the testing evolution of the mean 3D Dice for AdaptSegNet and our approach, evaluated every 5 epochs. In both datasets, training is very unstable for the adversarial approach. As a consequence, the performance can differ drastically depending on the number of training epochs and the stopping criterion. On the other hand, the proposed method shows a significantly better stability, smoothly converging during training.

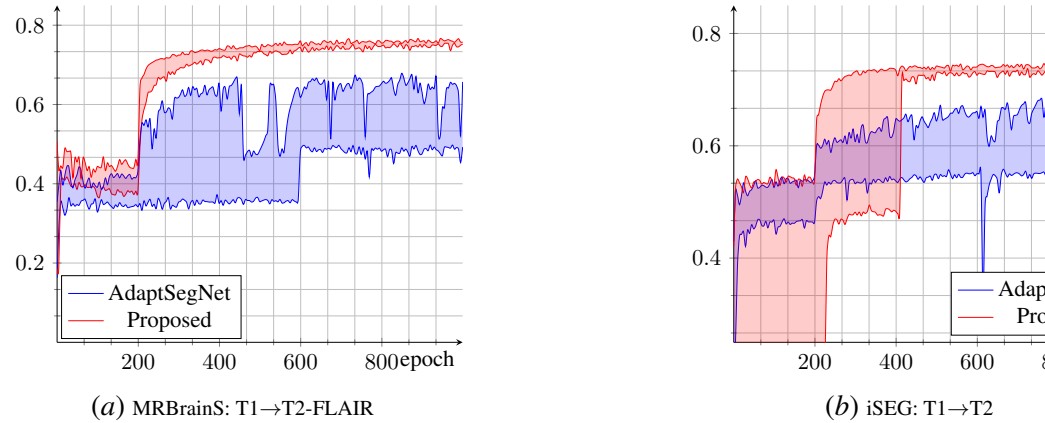

($a$) MRBrainS: T1→T2-FLAIR      ($b$) iSEG: T1→T2

Figure 3: Evolution of mean Dice coefficient over epochs. The minimum and maximum observed value over the three cross-validation runs is plotted, and the area in between is shaded.

**Kernel choice:** In addition to KL divergence, we conducted experiments with the squared Euclidean distance $\mathcal{D}(\mathbf{s}, \mathbf{s}') = \|\mathbf{s} - \mathbf{s}'\|^2$ and the negative Bhattacharyya kernel $\mathcal{D}(\mathbf{s}, \mathbf{s}') = -\sqrt{\mathbf{s}^T \mathbf{s}'}$, on both datasets. As shown in Table 3, the kernel choice has a negligible impact on the performances.

| $\mathcal{D}$ | Sq. Euclidean | Bhattacharyya | KL divergence |
|---|---|---|---|
| $\lambda$ | 0.1 | 0.1 | 0.01 |
| iSEG, T1 $\rightarrow$ T2 | $74.86 \pm 0.26$ | $74.81 \pm 0.49$ | $74.17 \pm 0.78$ |
| MRBrainS, T1 $\rightarrow$ T2 | $77.12 \pm 0.40$ | $77.96 \pm 0.36$ | $76.10 \pm 0.45$ |

Table 3: Mean Dice coefficients in percent when training with different distance functions $\mathcal{D}$.

**Impact of parameter $\lambda$:** We experimented with different value of parameter $\lambda$ to examine the sensitivity of the method with respect to the choice of this parameter. The results are reported in Table 4.

| $\lambda$ | 0.1 | 0.05 | 0.01 | 0.005 | 0.001 |
|---|---|---|---|---|---|
| iSEG, T1 $\rightarrow$ T2 | $74.70 \pm 0.46$ | $74.85 \pm 0.48$ | $74.17 \pm 0.78$ | $74.60 \pm 0.85$ | $72.68 \pm 0.51$ |
| iSEG, T2 $\rightarrow$ T1 | $78.37 \pm 0.40$ | $78.22 \pm 0.55$ | $77.89 \pm 1.15$ | $77.68 \pm 1.07$ | $77.29 \pm 0.64$ |
| MRBrainS, T1 $\rightarrow$ T2 | $77.07 \pm 0.59$ | $77.19 \pm 0.21$ | $76.10 \pm 0.45$ | $75.79 \pm 0.52$ | $71.21 \pm 0.71$ |
| MRBrainS, T2 $\rightarrow$ T1 | $82.39 \pm 0.29$ | $82.52 \pm 0.52$ | $82.43 \pm 0.50$ | $81.66 \pm 0.58$ | $78.96 \pm 0.21$ |

Table 4: Mean Dice coefficients in percent when training with different Lagrange parameters.

## 4. Conclusions

In this paper, we proposed a direct distribution matching approach for UDA in the context of semantic segmentation of medical images. Unlike adversarial approaches, our method matches the distributions from both domains with a single network, avoiding complex and unstable adversarial steps. It also leverages the contextual similarities of the output (label) spaces corresponding to pairs of images from different modalities but depicting the same structures, up to some geometric transformations, as is very common in medical imaging. Unlike natural images, this property is specific to multi-modal medical images and provides a very important structure prior for UDA. Adversarial approaches do not have a mechanism to account for such an important prior. As demonstrated in our experiments, directly matching output distributions has several benefits compared to adversarial learning: significantly superior performances and better training stability.

## Acknowledgments

Dr. Georg Pichler and Prof. Pablo Piantanida would like to acknowledge support for this project from the CNRS via the International Associated Laboratory (LIA) on Information, Learning and Control. The work of Prof. Pablo Piantanida was supported by the European Commission's Marie Sklodowska-Curie Actions (MSCA), through the Marie Sklodowska-Curie IF (H2020-MSCAIF-2017-EF-797805-STRUDEL).

Prof. Jose Dolz would like to thank NVIDIA for the donation of one TITAN V to support his research.

Some computations were made on the supercomputer "Helios" from Laval University, managed by Calcul Québec and Compute Canada. The operation of this supercomputer is funded by the Canada Foundation for Innovation (CFI), the ministère de l'Économie, de la science et de l'innovation du Québec (MESI) and the Fonds de recherche du Québec - Nature et technologies (FRQ-NT).

Part of this work was performed using HPC resources from the Mésocentre computing center of CentraleSupélec and École Normale Supérieure Paris-Saclay supported by CNRS and Région Île-de-France.

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
