# OpenReview forum: "On Direct Distribution Matching for Adapting Segmentation Networks"
_MIDL.io/2020/Conference — MIDL 2020_

### Official Review · AnonReviewer3 · 2020-03-10
**Interesting approach, but the setup of domain adaptation in this paper is quite different from the literature**

**Rating:** 2
**Confidence:** 4

**Summary:**

The paper proposes a domain adaptation approach that aims to directly minimize the discrepancy between source and target examples for adapting segmentation networks. The setup is to adapt brain segmentation across different MRI modalities. The results are shown to outperform standard adversarial domain adaptation approaches.

**Strengths:**

1. The paper is well-written and easy to understand.
2. The paper demonstrates that using paired source-target images are beneficial to domain adaptation.
3. The proposed method works well on MRBrainS.

**Weaknesses:**

1. The domain adaptation setup. Domain adaptation usually refers to the marginal distribution matching between p(s) and p(t). However, in this paper, the goal is to match the joint distribution p(s,t). Due to the additional information manifested through the joint distribution, i.e., paired data, it is easier to learn better representations across different domains. In essence, the effectiveness of this model is tightly coupled with the need for paired data.

2. The definition of domain gap. One strong assumption of this paper is that the domain shift is restricted to different MRI modalities *within the same scanner and the same patient*. However, domain shift can also present in different scanners or datasets which cannot be addressed by the proposed method. It is more valuable to address cross dataset domain shift than the type of domain shift presented in this paper.

3. Evaluation. It is not clear how the evaluation is carried out. It appears that all the source data are used in training, which raises the question of what kind of generalization do we want to evaluate. There could be two types of generalization: (i) generalizing an example x_i from the labeled source modality to the same example x_i in the target modality, and (ii) generalizing an example x_i from an unlabeled source modality to the target modality. Because of the shared label structure between domains, it will be more interesting to evaluate examples that were not trained in the source domain, i.e., evaluate on a hold-out set from the source/target.

4. Baseline. The setup of this paper is different from  (Tsai et al., 2018) due to the availability of the joint distribution p(s,t). It is not clear how the baseline (Tsai et al., 2018) is implemented in this paper. Is it based on marginal distribution matching where the source and target examples are shuffled, or does it have the same setup with the proposed approach where the source and target examples are always paired at the example level?

**Justification Of Rating:**

1. The definition of domain gap is restricted to paired data at the example level and does not consider cross-scanner or cross dataset domain shift.
2. Motivation and evaluation are not well justified.

**Paper Type:**

methodological development

**Questions To Address In The Rebuttal:**

Please address the questions in the "Weaknesses" section.

**Special Issue:**

no

---

> ### Author Response · Authors · 2020-03-28
> **Thank you very much for taking the time to review our paper.**
>
> Thank you very much for taking the time to review our paper. It seems to us that the wording in the paper caused some confusions. We will certainly address these issues in a revision. Find our individual answers below.
>
> 1. In essence this is correct. Our approach is designed to take advantage of the paired data samples. We do, however, believe that this situation is practically relevant and want to point out that adaptation in our proposed approach does not completely fail in case the alignment is imperfect. For details, we refer also to answer 1, given to reviewer 1. [TODO: can reviewers see other answers?]
>
> 2. The situation we want to address (see also answer 1 given to reviewer 1) is the setting where scans of the same patient but different scanners are available. As we did not find such data in publicly available datasets, we decided to use different modalities as a proxy. Certainly, applying the proposed approach to different kinds of domain shift will strength the results of this work.
>
> 3. Not all source domain data was utilized during training. The evaluation procedure is again outlined in detail in answer 5, given to reviewer 1. We will certainly clarify this in a revision. Evaluation was performed exactly as suggested. It was conducted on a hold-out set (consisting only of one sample) from the labeled source/target data.
>
> 4. We conducted the training of AdaptSegNet using the same pairings at example level. However, we found that permutation of the target domain samples, i.e., paring them up with the wrong source domain samples, does not impact the performance of AdaptSegNet. This is due to the training procedure used in AdaptSegNet.

---

> > ### Comment · AnonReviewer3 · 2020-03-30
> > **Thanks for the clarification**
> >
> > I am satisfied with the authors' feedback and would like to raise my rating to "3 weak accept".
> >
> > Please make sure to clarify the following issues in the revision:
> > 1. the domain adaptation setup in this paper, its motivation and how it differs from the literature;
> > 2. the training and evaluation procedure in terms of how the data is split;
> > 3. paired training with adversarial approaches is not as effective.
> >
> > Besides, I think this approach has a strong connection to self-training in that the trained source model provides supervisory signals to the target domain through discrepancy minimization.

---

### Official Review · AnonReviewer4 · 2020-03-10
**Good motivation and results**

**Rating:** 4
**Confidence:** 3
**Recommendation:** Poster

**Summary:**

This paper proposes a new Unsupervised Domain Adaptation loss for segmentation networks. The loss consists on the usual cross entropy between predictions and targets (on the labeled domain) and an additional loss based on density matching in the network’s output space, which is computed between inputs from different domains. The additional loss encourages inputs from different domains to produce the same output, up to geometric transformation of the input.

**Strengths:**

+ The proposed method is a simple addition to a standard supervised pipeline for training segmentation networks.
+ Two public benchmarks are used for evaluation.
+ The proposed idea is well motivated and explained throughout the paper, and backed with relevant experiments.

**Weaknesses:**

- The authors compare Mainly against AdaptSegNet (Tsai et al. 2018), which uses a different network architecture. It would have been interesting to see the same architecture used with the Density Matching loss, instead of the Adversarial Domain Adaptation loss in the original paper, to better compare both approaches.

**Justification Of Rating:**

This paper has a good related work overview leading to a detailed explanation of the proposed loss for unsupervised domain adaptation (UDA). The proposed technique is backed both theoretically and empirically.

**Paper Type:**

methodological development

**Special Issue:**

yes

---

> ### Author Response · Authors · 2020-03-28
> **Thank your for the excellent review.**
>
> Thank you very much for your excellent review.
> We agree, that more work on different segmentation network architectures would certainly be interesting. However, we decided to focus on the methodological aspect of domain adaptation by trying to avoid the need for a computationally expensive model, while facilitating a fair comparison. This was achieved by using the same segmentation network for both approaches.

---

### Official Review · AnonReviewer1 · 2020-03-12
**A domain adaptation method for particular application scenarios, more justification and evaluation are necessary**

**Rating:** 2
**Confidence:** 5

**Summary:**

This paper proposes a domain adaptation method for scenarios where the source and target data are paired and have identical ground truth. It minimizes the discrepancy, such as the KL divergence, between the prediction distribution of the source and target domain. Evaluation is conducted on paired and registered MRI data with two similar brain segmentation tasks.

**Strengths:**

- The paper is easy to read and well organized.
- The presented method is evaluated on public datasets.
- Ablation studies on training stability, kernel choice, and impact of hyper-parameter are conducted.

**Weaknesses:**

My major concerns about this paper is the general applicability of the proposed method as the defined domain-adaptation setting seems to have limited application scenarios. Also, the experimental evaluation is not comprehensive enough to support the efficacy of the proposed method.

**Detailed Comments:**

- Adding a figure illustrates the overview of the proposed method could help describe the work better.
- The source and target data are already aligned in both datasets. It would be great to see the effect of misalignment on the adaptation performance.
- It is unclear how the labeled and unlabeled scans in both datasets are split and used in the training and evaluation.
- Both datasets used in evaluation are for the WM/GM/CSF segmentation in the brain. To demonstrate the generalizability of the proposed method, a much different dataset and task would be better.


**Justification Of Rating:**

The proposed method may have limited application scenarios as the defined domain adaptation setting is relatively strict. Also, the experimental evaluation is not comprehensive enough to demonstrate the efficacy of the proposed method.

**Paper Type:**

methodological development

**Questions To Address In The Rebuttal:**

- The presented method is designed for cases where the source and target domains have paired data and identical ground truth. In such scenarios, does the target test data have paired source data? If so, we can directly apply the source model on the paired source data to obtain the prediction, which can also be the prediction of the target test data. If not, then the defined domain-adaptation setting seems to be a quite particular case as it requires that only part of the source training data has ground truth, only part of the source training data has paired target data, and the target test data doesn’t have paired source data. The authors are suggested to justify the necessity of domain adaptation in such scenarios
- Recently, there are many domain adaptation works proposed for domain alignment in image space, feature space, or output space. Comparison with more state-of-the-art domain adaptation methods is needed to validate the efficacy of the proposed method.

**Special Issue:**

no

---

> ### Author Response · Authors · 2020-03-28
> **Thank you for the detailed comments and critizism.**
>
> Thank you very much for the detailed review. We agree that additional evaluation of our approach would be desirable. However, suitable training data is scarce in publicly available datasets. We do believe that we found a good replacement, by using different acquisition protocols for source and target domain. Please allow us to address all points individually below.
>
> 1. The domain adaptation scenario is motivated by the following prototypical situation: After data acquisition, labeling, training and operation of a segmentation ML system, eventually the equipment needs to be phased out and new MRI equipment is purchased. The resulting domain shift between the two machines will severely degrade the performance of the segmentation system. In order to avoid the costly labeling step, which needs to be conducted by human experts, we propose a domain adaptation, which uses the existing labeled source data and images acquired from the same patient using the old (source domain) and new (target domain) equipment. For evaluation a few labeled target domain samples are needed, but no labeled target domain data is required for training.
> Unfortunately we were unable to find publicly available medical datasets that feature scans of the same patient, acquired with different equipment. Thus, we decided to use different modalities as a proxy. Considering that the different modalities needed to be aligned in a post-processing step in both considered datasets (as pointed out in the respective publications) made us confident that this is indeed a useful proxy. On the contrary, naively, one would even expect the domain shift to be larger between different modalities than between different equipment using the same modality.
>
> 2. We compare our approach to AdaptSegNet, which also allows for feature space, output space, or combined output and feature space alignment. As the paper reports an improvement of only one percentage point of the combined approach over the feature space approach, we opted for the latter to keep training simpler. Certainly, further comparisons would be interesting.
>
> 3. We will add a figure for illustration in a revision, if space constraints permit.
>
> 4. We did perform studies on the effect of disalignment by deliberately shuffling the target data scans, i.e., pairing them to the wrong source data scans, and subsequently using an affine registration algorithm to roughly align the images. Due to space constraints we did not include the table in the paper. As expected, we found severe degradation of our approach by 17.7 percentage points of mean DICE on average over all four modality/dataset combinations. However, it still outperformed the "no adaptation" approach by 17.9 percentage points. AdaptSegNet slightly outperformed our approach by 1.7 percentage points on average on this modified dataset. This suggests that, while our approach achieves comparable results to the state-of-the-art AdaptSegNet method when disalignment exists, it efficiently leverages pairs of images that can be aligned. We will include these numbers in a revision.
>
> 5. Both datasets, iSEG and MRBrainS, include unlabeled testing data and labeled training data, provided by the challenge organizers. The unlabeled testing data is used as the source/target pairs for calculating the second term in the loss function (1), while all but one scans (including all the modalities) of the labeled training data were used for the calculation of the first term of the loss function (1). The target domain data of these scans was not utilized. The remaining labeled scan was used for evaluation. We performed this training three times, using a different labeled scan for evaluation each time. The reported numbers are the mean over these three runs and their empirical standard deviation.
>
>   6. We agree that verification on a different task and dataset would beneficial, but suitable training data that features aligned source and target domain data is scarce in publicly available datasets. We are currently working on extending our method to other datasets for a more comprehensive evaluation. Nevertheless, the goal of this paper is to demonstrate the feasibility of the approach.

---

> > ### Comment · AnonReviewer1 · 2020-04-03
> > **The motivation is still not convincing enough**
> >
> > Thank the authors for the clarification, however, I am not convinced about the domain adaptation setup considered in this work and concerned about its applicability.
> >
> > The authors argue that the proposed domain adaptation is for "images acquired from the same patient using the old (source domain) and new equipment (target domain)", however, in such scenarios, the ground truth of source and target data may not be identical, which is required by the proposed method. As an example, in the longitudinal study of patients with brain lesion, the brain lesion may progress such that the brain lesion segmentation in the images acquired with the old and new equipment would change for the same patient.

---

> > > ### Author Response · Authors · 2020-04-03
> > > **There is a breadth of medical imaging problems where the source and target share a common output structure, up to some geometric transformation**
> > >
> > > We agree that our scenario may not be applicable to abnormalities that may undergo significant transformations, such as tumours and lesions. We are not claiming complete generality of our method. While tumours and lesions are important problems, they are applications where our assumption of a common output structure might be violated, unless the time between scans is kept short.  However, there is a wide range of organ segmentation problems, where the organs in the target and source domains are related by the same underlying structure, up to some simple geometric transformation. In these cases, one can leverage a long history of powerful registration algorithms in medical imaging, even when the source-target pair does not involve the same patient. This is precisely the message of our paper: in such frequent medical imaging scenarios, our simple distribution matching approach, along with registration, removes the need for complex and unstable adversarial methodologies. Our results clearly point to this, showing that, in the same setting, we obtain a substantially better performance than a state-of-the-art adversarial method, while stabilizing training.

---

> > > > ### Comment · AnonReviewer1 · 2020-04-06
> > > > **Thanks for the clarification**
> > > >
> > > > With the authors' further clarification, I feel the domain adaptation setting in this paper is reasonable with certain application value, and would like to raise my rating to "3 weak accept".
> > > >
> > > > The authors are suggested to make sure the following points are included in the revision:
> > > > - Clearly describe the motivation of the domain adaptation setting, the advantages of the proposed method for applicable scenarios and the limitations.
> > > > - The effect of misalignment on the adaptation performance.
> > > > - Clearly describe how the data are split and used in training and evaluation.

---

### Comment · Area_Chair1 · 2020-04-01
**Discussion**

Dear Reviewers,

Thank you R3 for providing feedback to authors' rebuttal. For the other two reviewers, could you check take some time to check the rebuttal and provide feedback? Thanks!

---

### Meta-Review · Area_Chair1 · 2020-04-07
**MetaReview of Paper45 by AreaChair1**

**Rating:** 4
**Recommendation For Accepted Papers:** Poster

**Metareview:**

All reviewers are convinced by the scientific value and evaluation results about this paper. A clear acceptance. The final version should clarify the motivation of the domain adaptation setting and its advantages, discuss the limitation of application scope, clarify data splits of training/evaluation, as pointed out by reviewers.

**Paper Type:**

both

**Special Issue:**

no

---

### Decision · Program_Chairs · 2020-04-11

Accept